# B-Cell Lymphomas Secrete Novel Inhibitory Molecules That Disrupt HLA Class II-Mediated CD4+ T-Cell Recognition

**DOI:** 10.3390/cells14151220

**Published:** 2025-08-07

**Authors:** Jason M. God, Shereen Amria, Christine A. Cameron, Lixia Zhang, Jennifer R. Bethard, Azizul Haque

**Affiliations:** 1Department of Pharmacology and Immunology, Medical University of South Carolina, 173 Ashley Avenue, Charleston, SC 29425, USA; godjasonm@gmail.com (J.M.G.); shereen.amria@gmail.com (S.A.); christinecameron9@icloud.com (C.A.C.); lixiazha@buffalo.edu (L.Z.); bethard@musc.edu (J.R.B.); 2Hollings Cancer Center, Medical University of South Carolina, 86 Jonathan Lucas Street, Charleston, SC 29425, USA; 3Department of Neurosurgery, Medical University of South Carolina, 96 Jonathan Lucas Street, Charleston, SC 29425, USA; 4Ralph H. Johnson Veterans Administration Medical Center, 109 Bee St, Charleston, SC 29401, USA

**Keywords:** B-cell lymphoma, HLA class II, lymphoma-associated molecules, dendritic cells, immune escape, CD4+ T cells, immunotherapy

## Abstract

B-cell lymphomas, including Burkitt lymphoma (BL), diffuse large B-cell lymphoma (DLBCL), and follicular lymphoma (FL), evade CD4+ T-cell immunity through novel HLA class II-associated immunosuppressive mechanisms. Despite expressing surface HLA-DR, these tumors fail to activate antigen-specific CD4+ T cells, independent of co-stimulation or PD-L1 checkpoint inhibition. We identified lymphoma-secreted factors that broadly disrupt HLA class II-mediated antigen presentation in both malignant B cells and dendritic cells (DCs), silencing T-cell responses. This inhibition is allele-independent (affecting DR1, DR4, DR7) but spares HLA class I-mediated CD8+ T-cell recognition, indicating a targeted immune evasion strategy. Biochemical and mass spectrometry (MALDI-MS) analyses revealed unique low-molecular-weight peptides (693–790 Da) in BL cells, absent in normal B cells, which may mediate this suppression. Functional fractionation confirmed bioactive inhibitory fractions in lymphoma lysates, further implicating tumor-intrinsic molecules in immune escape. These findings highlight a previously unrecognized axis of B-cell lymphoma immune evasion, where secreted factors disable HLA class II function across antigen-presenting cells. Therapeutically, neutralizing these immunosuppressive molecules could restore CD4+ T-cell surveillance and enhance immunotherapies in B-cell malignancies. This work underscores the importance of HLA class II dysfunction in lymphoma progression and identifies candidate targets for reversing immune suppression.

## 1. Introduction

B-cell lymphomas are cancers arising from malignant B lymphocytes, often characterized by immune evasion and dysregulated proliferation [1,2,3]. Burkitt lymphoma (BL), an aggressive subtype linked to MYC oncogene activation, is associated with Epstein–Barr virus (EBV) in endemic cases [4,5]. Despite its rapid growth, BL elicits an immune response, including cytotoxic T cells and NK cells. However, tumor-derived factors (e.g., soluble HLA molecules or exosomes) may impair dendritic cell (DC) function, hindering antigen presentation and T-cell activation. Understanding these immune escape mechanisms is critical for improving immunotherapies.

Lymphoid malignancies, including BL, diffuse large B-cell lymphoma (DLBCL), and follicular lymphoma (FL), exploit multiple mechanisms to evade immune surveillance, contributing to the progression of disease and induction of therapeutic resistance [6,7,8,9]. Effective antitumor immunity depends on the coordinated activation of tumor antigen (Ag)-specific CD8+ cytotoxic T lymphocytes (CTLs) and CD4+ helper T cells, which recognize antigens presented by HLA class I and II proteins, respectively [10,11,12,13]. While the role of CD8+ T cells in tumor control is well-established [14,15], emerging evidence highlights the important contribution of CD4+ helper T cells in sustaining durable antitumor responses through direct cytotoxicity, cytokine production, and support of CD8+ T-cell and B-cell function [16,17]. However, many B-cell malignancies, despite expressing HLA class II molecules, fail to optimally stimulate CD4+ T cells, suggesting an active immune evasion strategy.

BL, in particular, exhibits profound defects in antigen presentation by HLA class I-proteins, allowing escape from CD8+ T-cell recognition [18,19]. Intriguingly, we and others have demonstrated that BL cells also fail to stimulate CD4+ T cells via HLA class II proteins, even when presenting known antigenic peptides [20,21,22]. This defect is not due to a lack of expression of cell-surface HLA class II proteins, as BL and B-lymphoblastoid cell lines (B-LCLs) show comparable expression levels of HLA class II, invariant chain (Ii), HLA-DM, and HLA-DO proteins. Furthermore, attempts to rescue CD4+ T-cell activation through costimulatory signals (CD28 crosslinking) or B-cell receptor engagement (IgM crosslinking) prove ineffective [20,21]. Strikingly, this impairment extends to primary FL and DLBCL tumors, including EBV-negative cases, ruling out viral proteins such as EBV gp42 as a universal explanation. These findings suggest a broad, tumor-intrinsic defect in HLA class II antigen processing or presentation across B-cell malignancies.

Recent work has implicated soluble inhibitory factors in tumor-mediated immune suppression [23,24]. We discovered that culture supernatants from BL, FL, and DLBCL cells severely impair HLA class II antigen presentation by both B-LCLs and DCs [20,21]. This inhibition is not attributable to known immunosuppressive molecules (e.g., IL-4, TGF-*β*, IL-10, PD-L1, PGE2, IDO), suggesting the involvement of novel inhibitory factors. Studies using mass spectrometry (MALDI-MS) analysis of acid-eluted BL cell extracts revealed unique low-molecular-weight peaks absent in B-LCLs, providing the first biochemical evidence of lymphoma-derived immune disruptors. Further characterization of gel-fractionated BL proteins identified specific bioactive fractions that suppress CD4+ T-cell activation depending on the dosage, reinforcing the existence of a potent, yet unidentified, immunosuppressive mechanism.

The broader implications of this immune evasion strategy are significant. Since DCs are essential for priming antitumor T-cell responses [25,26,27], the ability of lymphoma-secreted factors to impair DC function suggested a systemic immunosuppressive effect within the tumor microenvironment. This could explain the poor clinical responses to immunotherapies (e.g., checkpoint inhibitors) in certain B-cell malignancies, despite adequate HLA class II expression. Moreover, the allele-independent nature of this inhibition (affecting HLA-DR1, DR4, and DR7 but sparing HLA-A2) points to a generalizable mechanism that could be targeted therapeutically.

Recent studies have also identified several previously unrecognized secretory molecules and vesicle-mediated pathways in B-cell lymphomas that contribute to tumor progression, immune evasion, and microenvironment remodeling [28,29]. Key among these is interleukin-10 (IL-10), which is secreted predominantly by activated B-cell-like DLBCL (ABC-DLBCL) and fosters an immunosuppressive environment that supports tumor growth, with high IL-10 levels correlating with poor prognosis and treatment resistance [30,31]. Malignant B cells also secrete lymphotoxin, particularly in follicular lymphoma and DLBCL, which reprograms stromal cells to create a tumor-supportive niche associated with disease persistence. Additionally, B cells can export peptide–MHC class II (pMHC II) complexes via exosomes, enabling antigen presentation without direct cell contact—an emerging mechanism whose clinical significance is still being defined [32]. These findings highlight novel biomarkers and potential therapeutic targets across B-cell lymphoma subtypes.

In this study we tested a hypothesis that B-cell lymphomas secrete previously unrecognized molecules that disrupt HLA class II antigen presentation, rendering tumor cells invisible to CD4+ T cells and fostering an immunosuppressive niche. This study combines functional T-cell assays, proteomic profiling, and biochemical fractionation to define the mechanistic basis of HLA class II protein dysfunction in BL cells, and to identify and characterize the inhibitory factor(s) responsible for the disruption of T-cell recognition of lymphoma cells. By elucidating this immune evasion pathway, our work may uncover new targets for immunotherapy and improve strategies to harness CD4+ T cells against malignant B-cell lymphomas.

## 2. Materials and Methods

### 2.1. Cell Lines

The human B-lymphoblastoid cell line (B-LCL) Frev which innately expresses HLA-DR (DRB1*0401) was cultured in complete IMDM (Mediatech, Manassas, VA, USA) medium with 10% fetal bovine serum, 50 U/mL penicillin, 50 μg/mL streptomycin, and 50 μm *β*-mercaptoethanol (Invitrogen, Grand Island, NY, USA) [33,34]. The human B-LCL line 6.16 and human Burkitt’s lymphoma (BL) cell lines Nalm-6 and Ramos were cultured in complete RPMI-1640 medium as described previously [35,36]. The 6.16 line is a sub-clone of the parental 6.1.6 bare lymphocyte syndrome-like line [20,36]. The Nalm-6, Ramos, and 6.16 cell lines were retrovirally transduced with the HLA-DR4 (DRB1*0401) allele [37]. The 6.16 line was further transfected with DM to generate the 6.16.DR4.DM line [36]. Surface HLA-DR4 expression was confirmed by flow cytometric analysis using the DR4-specific monoclonal antibody 359-F10 [34]. DCs expressing HLA-DR4 (FSDC.DR4) were also produced as described previously. The expression of surface HLA-DR4 was confirmed by flow cytometric analysis using the HLA-DR4-specific monoclonal antibody 359-F10. T-cell hybridoma line 2.18a recognizes Igκ residues 188–203 [38,39], and the 17.9 and 50.84.17 lines respond to human serum albumin (HSA) residue 64–76 K and influenza hemagglutinin HA-flu_307–319_ respectively [40]. These T-cell hybridoma lines were maintained in RPMI-1640 with 10% fetal bovine serum, 50 U/mL penicillin, 50 μg/mL streptomycin, and 50 μm *β*-mercaptoethanol (Invitrogen, Grand Island, NY, USA).

### 2.2. Peptide

Human IgG kappa (Igκ) immunodominant κ_188–203_ (sequence: KHKVYACEVTHQGLSS, HSA_64–76K_ (sequence: VKLVNEVTEFAKTK), and influenza hemagglutinin (HA-flu_307–319_ sequence: PKYVKQNTLKLAT) peptides were produced using Fmoc technology as described previously [21,37,39,40]. Peptide purity (>99%) and sequence were tested by reverse-phase HPLC purification and mass spectrometry. Peptides were then dissolved in PBS and stored at −20 °C until used.

### 2.3. Antigen Presentation Assays

B-LCL and BL were incubated with Igκ_188–203_, HSA_64–76K,_ or HA-flu_307–319_ synthetic peptide for 6 h to overnight at 37 °C in the appropriate cell culture media [39,41]. In cases of pretreatment, cells were treated with B-LCL or BL cell supernatants or cell extracts for 3 h before adding peptides. To prepare cell supernatants, cells (2 × 10^6^) were cultured in 1X AIMV serum-free media (Gibco, Grand Island, NY, USA) for 48 h. The supernatants obtained were collected and filtered using a 10 K Amicon Ultra Centrifugal Filter (Millipore, Burlington, MA, USA) in which supernatants were centrifuged at 2500 rpm for 20 min at 4 °C. Supernatants were then collected and stored at 4 °C. After incubation with the specific peptides, and supernatants or eluates, the cells were washed twice to remove residual peptides and cocultured with the specific T-cell hybridoma lines for 24 h [37,42]. For all assays, after coculture, the production of IL-2 was measured by ELISA [41]. Cell viability was tested by trypan blue cell counting at the end of the assay. The assays were repeated at least three times, and the data were presented as the means ± SEMs of at least three separate experiments.

### 2.4. Collection of Cell Eluates

To assess T-cell production of interleukin 2 (IL-2), B-lymphoblastoid lines and BL lines were first co-cocultured with one of the following: whole IgG κ, CLL synthetic peptide, or HSA peptide (0–20 µM) for 3–24 h at 37 °C. ELISA was then performed to quantify T-cell production of IL-2 in the co-cultures [15,37]. Triplicate assays were performed, and the standard errors calculated for triplicate wells within a single assay are reported. Further, acid eluates were collected from the following cell lines, which were washed 1× in pH 5.5 citrate phosphate buffer (CPB) and then shaken in CPB for 4 h: 6.16.DR4.DM, Ramos.DR4, Nalm-6.DR4. Following shaking of the cells in CPB, the cell suspensions were spun down and the eluates collected for further analysis.

### 2.5. Enzyme-Linked Immunosorbent Assay

Analysis of IL-2 and IFN γ in cell supernatants from triplicate wells in 96-well plates was performed by ELISA (R&D Systems, Minneapolis, MN, USA) following the manufacturer’s instructions [15,37]. Anti-IFN γ and IL-2 were purchased from R&D Systems and Sigma Aldrich (St. Louis, MO, USA), respectively. Triplicate assays were performed.

### 2.6. Protein Extraction and Digestion

Protein content was analyzed in the acid eluates of Nalm-6.DR4 and 6.16.DR4 cell lines, which were prepared by our lab as described previously [15]. Following concentration of the eluates, protein concentrations were measured and samples were run on a non-reducing gel which was then stained with Coomassie blue [15,16]. For protein extraction, gel plugs for bands of interest were excised, washed in Eppendorf tubes with 50 mM ammonium bicarbonate for 10 min, and destained with 25 mM ammonium bicarbonate in 50% acetonitrile for 15 min. Destaining was repeated 2×. Following destaining, each plug was dehydrated with 100% acetonitrile for 15 min and dried in a SpeedVac. Digestion was performed by covering each destained and dehydrated gel plug with proteomics-grade trypsin (Sigma) and incubating overnight at 37 °C. The digestion supernatant was collected in a clean and dry Eppendorf tube. Further extraction of peptides from the gel plugs was performed by washing 1× in 25 mM ammonium bicarbonate for 20 min followed by washing 3× in 5% formic acid and 50% acetonitrile for 20 min each. The supernatants from all 3 washes were collected, pooled, and dried in SpeedVac Vacuum Concentrators (Waltham, MA, USA) to ~1 µL.

Dried samples were reconstituted with 10 µL of a 0.1% trifluoroacetic concentrate with a C18 Ziptip (Millipore, Billerica, MA, USA) and then eluted with 0.1% trifluoroacetic acid, 50% acetonitrile, and 7.0 mg/mL α-cyano-4-hyrroxycinnamic acid onto the matrix-assisted laser desorption/ionization (MALDI) target.

### 2.7. Mass Spectrometric Analysis Matrix-Assisted Laser Desorption/Ionization Time of Flight/Time of Flight (MALDI-TOF/TOF)

MALDI-TOF/TOF analysis of the dried spots was performed. The previously prepared spots were completely dried, and the plate was loaded into an Applied Biosystems 4800 Proteomics Analyzer (Shimadzu Scientific Instruments, Columbia, MD, USA). Prior to analysis, external calibration of the system was performed following the manufacturer’s standards and protocols. Analysis of samples was performed via batch mode using 2000 shots per spectrum. Peptide mass maps were first acquired over the *m*/*z* range of 800–3500 in reflectron mode with a delayed extraction time optimized for *m*/*z* 2000 by averaging 2000 scans to locate peaks of peptide origin. Mass-spectrometry (MS)-MS analyses were performed on the next batch run to obtain sequence data on the 20 most abundant peaks from the MS analysis. Data from batch processing were exported into the GPS Explorer data-processing system for interpretation and identification. The MASCOT database-searching algorithm analyzed the data and summarized the results in report format.

Databases were searched using two missed cleavages and one differential modification of methionine oxidation, and the top 20 matches were reviewed and assigned confident protein identifications. Protein separation in acid eluates was also performed on non-reducing 10% polyacrylamide gels. Appropriate protein bands were excised and extracted by sonication in PBS on ice, and the resulting extracts were added to BL cells or B-LCL incubated with K_188–203_ peptide, which were then used in antigen presentation assays.

### 2.8. Statistical Analyses

Statistical analysis was performed on data from three independent experiments for each experimental group. One-way ANOVA and Student’s *t*-test were applied to determine the statistical significance of differences between experimental groups. This statistical analysis was performed by GraphPad software (9.5.1) (Boston, MA, USA), with values of *p* ≤ 0.05 considered as significant.

## 3. Results

### 3.1. Deficiencies in CD4+ T-Cell Recognition of B-Cell Lymphoma Associated with HLA Class II Molecule Presentation

BL and B-LCL expressed measurable levels of cell-surface HLA class II DR. However, to easily compare Ag presentation by these cells, we first expressed a common DR allele in several BL or B-LCL lines. BL (Nalm-6 and Ramos) and B-LCL (6.16.DM) were retrovirally transfected with the class II allele DRB1*0401 as we had multiple distinct T-cell hybridoma lines recognizing peptides restricted by DR4w4. Flow cytometric analysis of the transduced BL/B-LCL samples confirmed that they all expressed surface HLA-DR4 molecules [20,43]. DR4^+^ BL cells (Nalm-6.DR4 and Ramos.DR4) or B-LCL (6.16.DR4.DM) were then sorted by flow, matched for surface DR4 expression, and incubated with the IgG kappa 188–203 peptide (kI) for 6 h. After incubation, cells were washed and cocultured with the peptide-specific T-cell hybridoma line (2.18a), and the supernatant was collected as we have previously described in the methods. The T-cell production of IL-2 by these or control-treated B-LCL was measured by ELISA. The production of IL-2 was calculated and shown in mean pg/mL by comparison with data obtained from a standard curve generated from recombinant IL-2 (R&D System). These data indicated that BL cells such as Nalm-6.DR4 and Ramos.DR4 did not stimulate the CD4+ T cells responsive to a distinct HLA class II-restricted epitope (k_188–203_) (Figure 1A). Following incubation with the IgG kappa 188–203 peptide (kI), these cells were cocultured with kI-peptide-specific human CD4+ T cells. The production of IFN-γ was measured by ELISA (R&D). These data suggest that, like IL-2, decreased IFN-γ production was recorded (Figure 1B). Overall, BL cells such as Nalm-6.DR4 and Ramos.DR4 were unable to optimally stimulate CD4+ T cells via the HLA class II pathway.

Studies suggest that many transformed cells secrete proteases/factors which may destroy peptides [44,45,46], thus disrupting their ability to display peptides or epitopes bound to HLA class II proteins. To test whether BL cells might be secreting factor(s) into the media which disrupt class II protein function, B-LCL (6.16.DR4.DM) cells were incubated with BL (Nalm-6 cells Nalm-6.DR4)- and B-LCL (6.16.DR4.DM)-conditioned media followed by the addition of synthetic k_188–203_ peptide. T-cell activation by these or control-treated B-LCL was measured by ELISA. Conditioned media from Nalm-6 cells decreased peptide presentation in B-LCL 6.16.DR4.DM cells (Figure 1C). Cell viability was not significantly altered (Figure 1D). These data indicated that shed BL molecules diminished CD4+ T-cell activation via the HLA class II pathway.

### 3.2. Disruption of CD4+ T-Cell Activation and MHC Specificity of BL-Associated Inhibitory Molecules

We examined whether the inhibitory effect is restricted to HLA-DR4 or other alleles. Our study suggests that the HLA-DR1-restricted antigen presentation is also perturbed by BL-derived molecules (Table 1). Briefly, BL (Nalm-6)-eluted materials were neutralized, passed through 10 K cut-off filters, and the retained fractions were concentrated to 1% of their total volume. Their protein concentration was determined using a standard assay routinely used in the laboratory. Frev cells that constitutively express HLA-DR1 were incubated with the kI, HAS, HA-flu, EBNA1, or survivin peptide (10 μM) plus Nalm-6 or Frev eluate (50 µg/mL) for 6 h. After incubation, the cells were washed and cocultured with their peptide-specific T-cell hybridoma for 24 h. T-cell production of IL-2 was measured by ELISA. Data obtained suggest that the Nalm-6-derived molecules perturbed peptide presentation by B-LCL (Frev) (Table 1, Appendix A). These data suggest that the Nalm-6 tumor-derived factor(s) disrupts DR7/DR4/DR1-mediated CD4+ T-cell responses. We also tested a class I allele (HLA-A2), which showed that the BL-eluted molecules did not inhibit HLA-A2 (class I)-restricted presentation of peptides and CD8+ T-cell responses (Table 1, Appendix A), suggesting that the inhibitory activity may depend on the class II allele. DR4- and DR7-restricted Ag presentation assays were also performed using primary CD4+ T cells. These studies showed a similar pattern as shown in Table 2. A pilot study also showed that the BL eluates did not alter HLA-A2 (class I)-restricted peptide presentation or CD8+ T-cell recognition (Table 1).

### 3.3. Mass Spectrometry of BL-Eluted Molecules Showed Different Spectra from BL Cells

We also subjected aliquots of the Nalm-6.DR4 eluate to MALDI mass spectrophotometry analysis to identify candidates with inhibitory activity. These studies using MALDI analysis showed that several new peaks representing low-mass species were only present in BL-eluted samples (Figure 2) and not B-LCL, suggesting that BL cells may produce inhibitory molecules that alter HLA class II presentation.

### 3.4. BL Shed Molecules Dose-Dependently Disrupt HLA Class II Ag Presentation

To examine the sensitivity of BL eluate, we cultured two B-LCL (Frev and 6.16.DR4.DM) and two BL cell lines (Nalm-6 and Ramos), and eluted BL/B-LCL-derived molecules as described above. These samples were passed through 10 K cut-off filters, and the retained fractions were concentrated to 1% of their total volume. The samples were processed for protein determination using a standard assay routinely used in the laboratory. Frev cells that innately express HLA-DR4 were then incubated with the kI peptide (10 µM) plus B-LCL/BL eluates (0–100 µg/mL) for 6 h. After incubation, cells were washed and cocultured with k188–203 peptide-specific T cells for 24 h. T-cell production of IL-2 was measured as described. These data showed that the addition of 25 µg/mL of crude extract from Nalm-6/Ramos cells significantly inhibited the HLA class II-mediated kI peptide presentation by B-LCL (Figure 3). Thus, BL-derived molecules may disrupt Ag presentation and immune recognition of malignant B cells.

### 3.5. BL Shed Molecules Also Inhibit HLA Class II Ag Presentation by DC

A number of studies suggest that APCs like DCs are potent APCs which influence HLA class II antigen presentation and the induction of antitumor immune response. To investigate whether shed B-cell lymphoma molecules inhibit HLA class II Ag presentation by DCs, we generated an HLA-DR4-expressing DC line, FSDC.DR4. FSDC.DR4 cells were then treated with Nalm-6 eluates for 3 h before adding the kI peptide for 6 h. A control buffer and the 6.16 eluate were used as controls. After incubation, cells were washed and cocultured with the kI peptide-specific T-cell hybridoma line 2.18a. The production of IL-2 by the T-cell hybridoma line was quantitated by ELISA, as previously described. The results obtained showed that eluates from Nalm-6 cells significantly decreased the presentation of the kI peptide by FSDC.DR4 cells (Figure 4).

### 3.6. Pools of Protein Bands Exhibit Inhibitory Activity

BL-eluted cellular extracts were concentrated and protein concentrations were quantitated and run on a non-reducing gel. Fourteen visible bands were observed when the gel was stained with Coomassie blue (Figure 5A). Gel plugs 4/5, 8/9, 12/13 were then excised and eluted in DDH2O, passed through a 10 kD filter, and the retained fractions were tested for inhibitory activity. These studies were unable to detect any significant reduction in CD4+ T-cell responses with these pooled proteins (Figure 5B). Band 10 was tested separately as it contained a sufficient amount of protein (needed for our assay), but it did not show any inhibitory activity. We then focused on the remaining bands (1/2/3/6/7/11 and 14), which were excised, pooled, and had their antigen presentation function tested as described. The data showed the significant inhibitory activity of these components in Ag presentation and activation CD4+ T-cell responses (Figure 5B). The Coomassie-blue-stained gel also showed that bands 10/11 and 12/13 were thicker in Frev cells than those of Nalm-6 cells (Figure 5A). However, these molecules did not have a significant effect on HLA class II protein function. These data suggest that one or more of the BL-derived molecules (1/2/3/6/7/11/14) perturb the activation of T cells via the HLA class II pathway.

### 3.7. MALDI Mass Spectroscopy and Amino Acid Sequencing Detected Protein Bands

We then analyzed these gel plugs using 4800 Plus MALDI-TOF-TOF at our mass spectroscopy facility. Data from mass spectroscopy and sequence analyses showed that bands 1, 3, 7, and 10 are known proteins (Table 2). Although parasitic profilins and CNS aldolase A can induce immune responses and Helminths peroxiredoxin 1 can reduce stress damage from hydrogen peroxide, they do not have a known inhibitory activity on Ag presentation. These studies using MALDI-TOF-TOF analyses showed that (i) protein bands 2 and 6 contain previously unknown proteins, (ii) band 11 contains muscle pyruvate kinase and an unknown/modified protein product, (iii) band 14 contains heat shock protein 90 (hsp90) with an unknown/modified protein product. It is unlikely that the immune inhibitory function resides within muscle pyruvate kinase, profilin, peroxiredoxin 1, or Aldoase A, but this can be tested in the future. To date, hsp90 by itself or in association with modified proteins has only been shown to augment antigen presentation by HLA class I and class II proteins. However, since most biological pathways provide mechanisms to both boost a protein’s activity and suppress it, it is feasible that these studies could identify previously unknown endogenous molecule(s) that may hinder hsp90’s chaperone of peptides to MHC class II molecules and thus suppress Ag presentation.

## 4. Discussion

This study reveals a critical immune evasion mechanism in B-cell lymphoma (BL), where tumor-derived factors impair CD4+ T-cell recognition by disrupting HLA class II-mediated antigen presentation. While BL cells express surface HLA-DR molecules, they fail to optimally activate CD4+ T cells specific to the IgG kappa 188–203 peptide (kI), as evidenced by significantly reduced IL-2 and IFN-γ production. This defect is not due to a lack of T-cell responsiveness but rather stems from BL-secreted inhibitory factors that interfere with HLA class II function, a finding with broad implications for tumor immunology and immunotherapy.

It is well-established that tumors employ multiple strategies to evade T-cell surveillance, including (a) the downregulation of HLA class I and class II proteins (e.g., in melanoma, glioblastoma) [47,48]; (b) altered peptide processing via defects in proteasomes, TAP transporters, or cathepsins [49,50], (3) the secretion of immunosuppressive factors (e.g., TGF-*β*, PGE2, IDO) that inhibit T-cell function [51,52]. However, the direct inhibition of HLA-II antigen presentation by secreted tumor factors is a less understood mechanism. Our data demonstrates that BL cells release soluble mediators that broadly suppress HLA-DR4, DR7, and DR1-restricted CD4+ T-cell responses, while sparing HLA-A2-restricted CD8+ T-cell recognition. This suggests a class-II-specific inhibitory pathway, distinct from global MHC downregulation.

In this study, we have identified BL-derived soluble inhibitors from conditioned media from Nalm-6 BL cells which suppressed B-LCL antigen presentation, indicating extrinsic disruption rather than intrinsic BL defects. Mass spectrometry revealed unique low-mass species in BL eluates, absent in B-LCL, suggesting tumor-specific factors. Specifically, this study uncovered dose-dependent and allele-specific suppression. Interestingly, even low concentrations (25 µg/mL) of BL extracts inhibited HLA class II presentation, highlighting potent biological activity. The preservation of HLA-A2 function implies a selective mechanism, possibly targeting (a) HLA class II peptide loading (e.g., blocking HLA-DM or invariant chain interactions) or (b) peptide stability (e.g., via proteases or chaperone interference), which remains to be tested.

DCs are very important for initiating antitumor immune responses in the context of HLA class II-mediated antigen presentation to CD4+ T cells [53,54,55]. However, our findings demonstrate that BL-derived soluble molecules (e.g., Nalm-6 eluates) significantly inhibit this process, as evidenced by the reduced production of IL-2 from T-cell hybridoma lines upon coculture with peptide-loaded DCs. This suppression may occur through downregulation of HLA class II or costimulatory molecules, induction of tolerogenic DC phenotypes, or disruption of antigen processing mechanisms, consistent with other B-cell malignancies like CLL, where tumor exosomes impair DC function. Clinically, these results highlight an immune evasion strategy employed by BL and suggest that restoring DC function through ex vivo DC vaccines, checkpoint inhibitors, or targeting immunosuppressive pathways could enhance immunotherapy efficacy.

Fractionation studies identified protein bands 1, 2, 3, 6, 7, 11, and 14 as key suppressors. MALDI-TOF-TOF detected hsp90, pyruvate kinase, and unknown/modified proteins. Intriguingly, Hsp90 typically enhances antigen presentation [56,57], but our data suggest that a modified or antagonistic form may exist in BL. The unknown proteins in bands 2, 6, and 14 could represent novel immune suppressors, meriting further characterization.

In B-cell lymphomas, particularly in aggressive subtypes such as primary mediastinal B-cell lymphoma (PMBCL), classical Hodgkin lymphoma, and subsets of DLBCL, PD-L1 is frequently upregulated on tumor cells and/or surrounding immune cells [58,59]. This upregulation leads to the engagement of the PD-1 receptor on T cells, inhibiting their activation, proliferation, and cytotoxic function. As a result, PD-L1 contributes directly to the establishment and maintenance of an immunosuppressive niche, wherein T-cell exhaustion, reduced interferon-γ production, and impaired antitumor responses are common. The upregulation of PD-L1 can be driven by genetic mechanisms, viral infection, or chronic inflammatory signaling [60]. This immune evasion strategy supports lymphoma cell survival and has been associated with poor prognosis, particularly when PD-L1 is coexpressed with other immunosuppressive factors like IL-10.

Many other lymphomas and hematopoietic tumors also secrete immunosuppressive and tumor-supportive molecules, which often correlate with poor prognosis [61,62]. Classical Hodgkin lymphoma secretes IL-10, TGF-*β*, and chemokines that attract regulatory T cells and suppress cytotoxic immunity, while also overexpressing PD-L1 factors linked to immune evasion and adverse outcomes [63]. Multiple myeloma secretes IL-6, VEGF, and BAFF, promoting tumor growth, angiogenesis, and immune suppression, with high levels associated with advanced disease and reduced survival [64]. T-cell lymphomas such as AITL and PTCL release cytokines like IL-10 and CXCL13, fostering a suppressive and angiogenic environment that drives aggressive behavior [65,66]. Chronic lymphocytic leukemia also secretes IL-10 and TGF-*β* and relies on stromal interactions to support tumor survival [67]; these features are associated with disease progression and therapy resistance. Overall, secretory activity across hematologic malignancies often contributes to immune evasion and worsened clinical outcomes.

The identification of BL-derived inhibitory factors opens promising translational avenues. Screening patient sera for these molecules could serve as a diagnostic tool to assess immune evasion status and personalize immunotherapy approaches. If validated, targeted strategies such as blocking antibodies against modified hsp90 or small-molecule inhibitors could restore CD4+ T-cell function. Given the limited capacity of immune checkpoint inhibitors (e.g., anti-PD-1) in BL, neutralizing these immunosuppressive factors may synergize with existing therapies to enhance T-cell responses. Furthermore, this mechanism may extend beyond BL to other HLA class II-positive malignancies, including Hodgkin lymphoma and certain carcinomas, suggesting broad relevance for therapeutic intervention. Future studies should focus on isolating these factors, testing neutralization in preclinical models, and exploring combinatorial regimens to overcome immune resistance.

In conclusion, our study uncovers a novel immune evasion strategy in BL, where tumor-secreted factors selectively disrupt HLA class II antigen presentation. By identifying candidate inhibitory proteins and demonstrating their functional impact, we provide a foundation for targeted interventions to restore antitumor immunity. Future work should prioritize purification of these factors, validation in primary patient samples, and preclinical testing of neutralizing agents, an innovative approach to overcoming immune resistance in B-cell malignancies.

## 5. Conclusions

This study reveals a critical immune evasion mechanism in B-cell lymphoma (BL), where tumor-derived factors impair CD4+ T-cell recognition by disrupting HLA class II-mediated antigen presentation. Despite expressing surface HLA-DR molecules, BL cells failed to stimulate CD4+ T cells specific to the IgG kappa 188–203 peptide, as shown by significantly reduced IL-2 and IFN-γ production. This defect was not due to intrinsic T-cell dysfunction but rather resulted from inhibitory molecules secreted by BL cells, which broadly suppressed HLA-DR4, DR7, and DR1-restricted CD4+ T-cell responses while sparing HLA-A2-restricted CD8+ T-cell recognition. Mass spectrometry and biochemical fractionation identified unique protein bands in BL eluates, suggesting tumor-specific factors that interfere with antigen presentation.

The inhibitory activity extended beyond direct tumor–T-cell interactions, as BL-derived molecules also impaired antigen presentation by DCs, a key cell type for initiating antitumor immunity. Fractionation studies pinpointed specific protein bands with immunosuppressive properties, some of which contained known proteins like hsp90 and pyruvate kinase alongside unknown or modified species. While hsp90 typically enhances antigen presentation, its modified form or associated proteins in BL may instead suppress immune recognition, highlighting a potential novel immune evasion strategy.

These findings open new avenues for therapeutic intervention in BL and other HLA class II-expressing malignancies. Future studies should focus on isolating the exact inhibitory factors, defining their mechanisms of action, and developing strategies to neutralize their effects. Restoring effective CD4+ T-cell responses could enhance immunotherapy efficacy, offering a promising approach to overcoming immune resistance in B-cell lymphomas. By targeting these tumor-derived suppressors, it may be possible to improve immune surveillance and clinical outcomes for patients.

## 6. Limitations

This study identifies key deficiencies in HLA class II protein-mediated antigen presentation by B-cell lymphoma cells and implicates tumor-derived factors in disrupting CD4+ T-cell activation. However, several limitations warrant consideration. First, while the inhibitory effect was specific to HLA class II alleles (DR1, DR4, DR7) and spared HLA class I (A2)-restricted presentation, the precise molecular mechanism remains unresolved. It is unclear whether BL-derived factors interfere with peptide loading, class II stability, or direct T-cell engagement. Further mechanistic studies, such as examining class II-peptide complex formation or intracellular trafficking, could clarify this selectivity.

A second limitation lies in the incomplete characterization of the inhibitory molecules. Mass spectrometry detected known proteins (e.g., hsp90, pyruvate kinase) without established roles in suppressing antigen presentation, suggesting that the active components may be post-translationally modified, part of undetected protein complexes, or present at levels below the assay’s sensitivity. The pooled gel bands (1/2/3/6/7/11/14) showed activity, but individual contributions were not parsed, leaving the key mediator(s) unidentified. The reliance on in vitro models, T-cell hybridomas, and immortalized B-LCLs also limits physiological relevance. While primary CD4+ T cells showed similar trends in experiments, patient-derived BL cells and tumor microenvironment conditions might reveal additional layers of regulation. Moreover, dose-dependent inhibition was demonstrated in controlled settings, but the physiological concentration of these factors in BL patients remains unclear.

Finally, the study examined a limited subset of BL cell lines (Nalm-6, Ramos) and HLA alleles. Whether these defects extend to other BL subtypes or class II variants (e.g., DP, DQ) is unknown. Broader profiling could determine if immune evasion via HLA class II disruption is a hallmark of BL or context-dependent. Despite these gaps, our findings provide a foundation for investigating how tumors exploit HLA class II pathways to evade immunity and highlights potential targets for therapeutic intervention.

## Figures and Tables

**Figure 1 cells-14-01220-f001:**
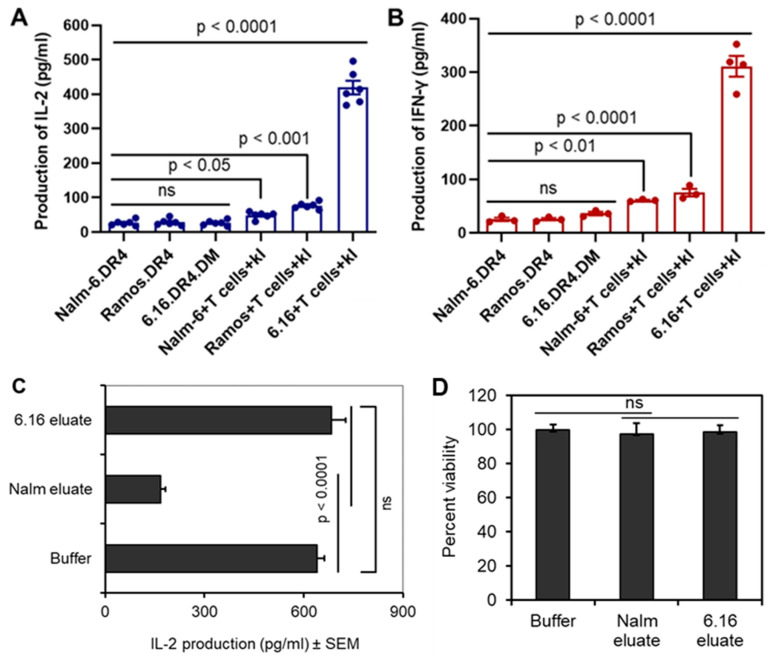
BL (Burkitt lymphoma) cells are unable to functionally present antigenic peptides to stimulate CD4+ T cells. (**A**) B-LCL (6.16.DR4.DM) and BL (Nalm-6.DR4, Ramos.DR4) lines were incubated with synthetic Ig kappa 188–203 peptide (kI) for 6 h, washed, and cocultured with the peptide-specific CD4+ T-cell hybridoma for 24 h. PBMCs from DR4+ve healthy individuals were repeatedly stimulated with Ig kappa 188–203 peptide (kI) to raise peptide-specific T-cell lines as described in the Materials and Methods. T-cell-secreted IL-2 was measured by ELISA. Data are the means +/− SEM of triplicate samples. (**B**) 6.16.DR4.DM, Nalm-6.DR4, and Ramos.DR4 cell lines were also incubated with kI peptide-specific human CD4+ T-cell lines in a 96-well plate for 48 h. T-cell-secreted IFN-γ was measured by ELISA. Data are the means +/− SEM of triplicate samples. (**C**) Cell supernatants were prepared from a BL cell line (Nalm-6) and a B-LCL line (6.16) as mentioned in the Materials and Methods. The B-LCL line Frev was incubated with the kI 188–203 peptide in the presence of control buffer or eluates from 6.16 and Nalm-6 cells. Antigen presentation assays were performed as described, and T-cell-secreted IL-2 was measured by ELISA. Data obtained are mean IL-2 production (pg/mL) +/− SEM of triplicate wells. (**D**) Cell viability was tested by trypan blue counting assay as described. ns = Not statistically significant. Statistical analyses were performed by Student’s *t*-test.

**Figure 2 cells-14-01220-f002:**
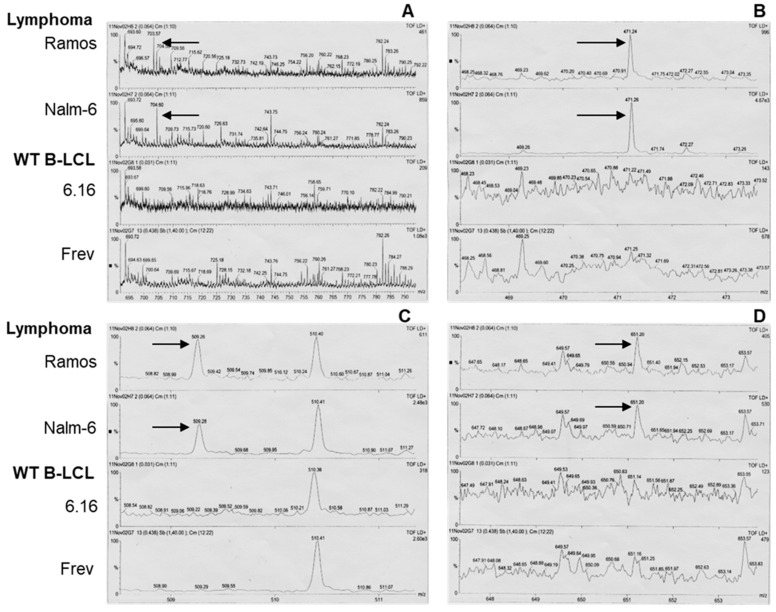
Partial expanded view of MALDI mass spectra of acidic-buffer-eluted samples from BL and wild-type B-LCL. Acid elusions (buffer pH 5.5) from two BL lines (Ramos and Nalm-6) and two wild-type B-LCLs (Frev and 6.16) were analyzed by MALDI. Arrows indicate unique spectral peaks consistently found only in BL cell lines. Expanded views of mass spectra at (**A**) 693.72–790.23, (**B**) 468.25–473.57, (**C**) 508.54–511.27, and (**D**) 647.65–653.57 daltons are shown.

**Figure 3 cells-14-01220-f003:**
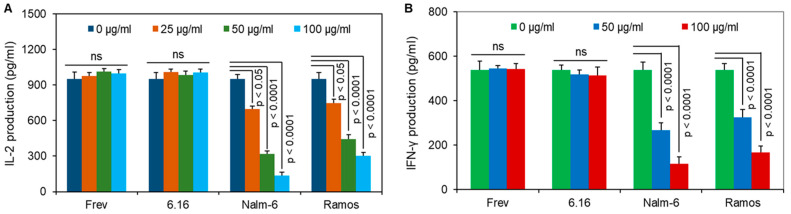
Effects of different doses of acid eluates on kI peptide presentation by a wild-type B-LCL, Frev. Acid elutions obtained from Frev, 6.16, Nalm-6, and Ramos cells were concentrated to approximately 1% of total volume. (**A**) The B-LCL line, Frev, was incubated with the kI peptide plus different doses (0–100 μg/mL) of acid eluates followed by a T-cell assay using the specific hybridoma line and quantitation of IL-2. (**B**) The B-LCL line, Frev, was incubated with the kI peptide plus different doses (0–100 μg/mL) of acid eluates followed by coculture with the peptide-specific human CD4+ T-cell line. The production of IFN-γ was measured by ELISA. The data suggests that the presentation of the kI peptide is significantly inhibited by the addition of Nalm-6 and Ramos eluates, and the inhibition of IL-2 or IFN-γ production is dose dependent. Statistical analyses were performed by Student’s *t*-test. ns = not statistically significant.

**Figure 4 cells-14-01220-f004:**
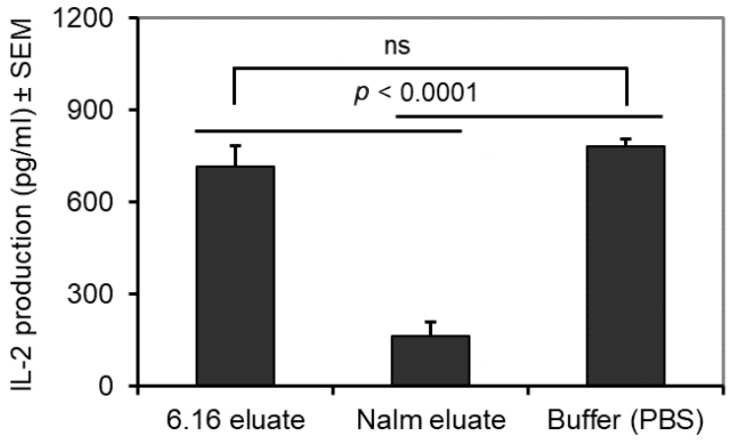
The acid eluate obtained from BL line decreased the functional presentation of kI peptide by DCs in the context of HLA-DR4. Cell eluates from a BL cell line (Nalm-6) and a B-LCL line (6.16) were prepared as described in the Materials and Methods. The DC line (FSDC.DR4) was incubated with the kI peptide in the presence of a control buffer or eluates from 6.16 and Nalm-6 cells. T-cell assays were performed as described, and the production of IL-2 was measured by ELISA. The data are the mean IL-2 production (pg/mL) ± SEM of triplicate wells. Statistical analyses were performed by Student’s *t*-test (*p* < 0.0001). ns = not statistically significant.

**Figure 5 cells-14-01220-f005:**
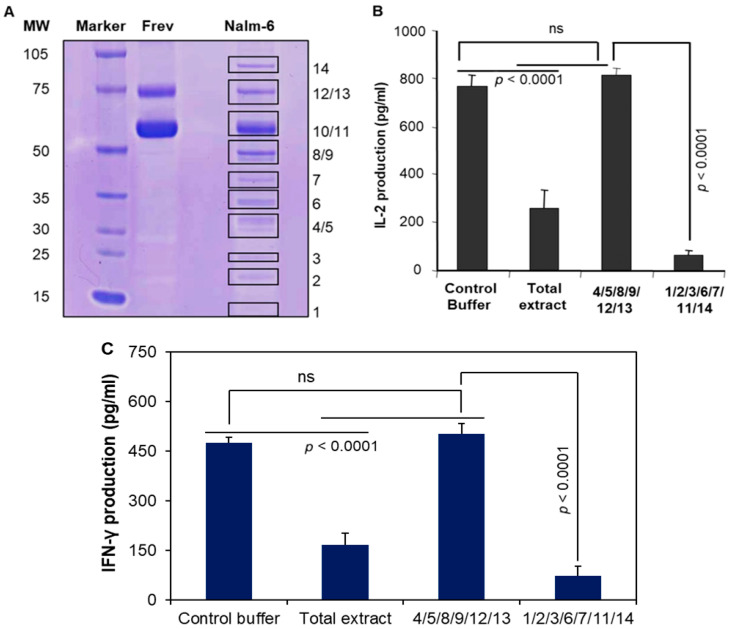
The effects of BL-associated molecules on the inhibition of antigen presentation via the HLA class II antigen presentation pathway. (**A**) B-LCL (Frev) and BL (Nalm-6) extracts (100 μg proteins) were separated on a standard non-reducing gel (4–12%) and stained with Coomassie blue. (**B**) The effects of gel-eluted proteins on kI peptide presentation by Frev B-LCL. BL Nalm-6 extract (500 μg proteins) was also run on a separate gel and stained with Coomassie blue. Then the 4/5/8/9/12/13 or 1/2/3/6/7/11/14 bands were excised, pooled, sonicated, and eluted in DDH2O. The extracts were passed through a 10 K filter, and the retained fractions were analyzed for protein determination and tested for inhibitory activity in T-cell assays as described. (**C**) The extracts were also tested in a T-cell assay using peptide-specific human CD4+ T cells, and the production of IFN-γ was quantitated by ELISA. Data suggest that the presentation of the kI peptide is markedly inhibited by the addition of either the total extract or the 1/2/3/6/7/11/14 gel eluates. Statistical analysis was performed by one-way ANOVA. Band 10 was tested separately and did not influence HLA class II Ag presentation.

**Table 1 cells-14-01220-t001:** Effects of BL-associated inhibitory molecules on Ag presentation by HLA class I and class II molecules. BL-derived molecules (50 µg/mL) were incubated with either HLA-DR4, HLA-DR7, or DR1 expressing B-cells followed by washing and co-culture with the HLA/peptide specific T cell hybridoma lines. T cell production of IL-2 was quantitated by ELISA. B-cells expressing HLA-A2 were cocultured with human CD8+ T cells, followed by measurement of IFN-γ in the culture supernatant. Data suggest that BL-eluted molecules disrupt DR4/DR7/DR1-restricted CD4+ T cell responses, but not HLA-A2-restricted presentation.

HLA-Restriction and Antigenic Peptide Presentation	Inhibition of T Cell Responses
HLA-DR4-restricted presentation of Igκ_188–203_ and Igκ_145–2159_ epitopes from whole IgGκ	Yes
HLA-DR4-restricted presentation of κ_188–203_ and κ_145–159_ synthetic peptides	Yes
HLA-DR4-restricted presentation of HSA_64–76K_ epitope from whole HSA	Yes
HLA-DR4-restricted presentation of HSA_64–76K_ synthetic peptide	Yes
HLA-DR4-restricted presentation of HA-flu_306–319_ peptide	Yes
HLA-DR7-restricted presentation of EBNA1_482–496_ peptide	Yes
HLA-DR7-restricted presentation of Survivin_20–34_ peptide	Yes
HLA-DR1-restricted presentation of Igκ_188–203_ epitope from whole IgGκ	Yes
HLA-DR1-restricted presentation of κ_188–203_ synthetic peptides	Yes
HLA class I (HLA-A2)-restricted Ag presentation	No

**Table 2 cells-14-01220-t002:** Biochemical analysis of gel plugs obtained from BL (Nalm-6) pH 5.5 extract. Gel plugs were excised and analyzed by MALDI TOF-TOF as described in the methods. NA = Not analyzed by MALDI TOF-TOF.

Bands	Known/Unknown Protein Products
1	Profilin 1
2	Unknown/modified protein
3	Peroxiredoxin 1
4/5	NA
6	Unknown/modified protein
7	Aldolase A
8/9	NA
10	Heat shock protein 60
11	Muscle pyruvate kinase, chain A + unknown/modified protein product
12/13	NA
14	Heat shock protein 90 + unknown/modified protein product

## Data Availability

The data used to support the findings of this manuscript are available from the corresponding authors upon reasonable written request after the publication.

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
