# Peer review of "B-Cell Lymphomas Secrete Novel Inhibitory Molecules That Disrupt HLA Class II-Mediated CD4+ T-Cell Recognition"

_cells, 2025, doi:10.3390/cells14151220_

Round 1
Reviewer 1 Report
Comments and Suggestions for Authors
The manuscript by God et al. describes a study uncovering a novel mechanism of immune evasion in B-cell lymphomas, where tumor-secreted factors impair HLA class II–mediated CD4⁺ T cell activation. Using functional assays and mass spectrometry, the authors identify candidate peptides involved in this process, paving the way for therapeutic intervention. However, the manuscript would benefit from clarification or improvement of a few points, as outlined below:
Major Points:
-In Section 2.3, the authors describe treating cells with conditioned media for 3h prior to antigen pulsing, followed by overnight incubation. It would strengthen the study to investigate whether the observed reduction in antigen presentation is due to HLA downregulation, impaired antigen uptake, or antigen retention—mechanisms already suggested in the second paragraph of the Discussion. It is also essential to demonstrate that the treatment does not compromise cell viability.
-In Figures 1, 3, and 5, consider including additional readouts of T cell responses, such as IFN-γ production, activation markers (e.g., CD69, CD25), or proliferation assays, to more comprehensively assess functional responses.
-Table 1 should include quantitative data and ideally converted into graphical format for clearer visualization.
-Figure 3 currently lacks statistical annotations. Please include appropriate statistical tests and significance indicators.
-Please clarify the technical rationale for the different incubation times used in the co-culture experiments described in Sections 3.1 (in details in 2.3) and 3.5, as this might affect comparability.
-Please indicate the statistical tests used in each figure legend, to ensure reproducibility and transparency.
Minor Points:
-In lines 88–89, there is an open but unclosed bracket, which should be corrected for clarity.
-In line 209, the word "conformed" appears to be a typo and should likely read "confirmed."
-In line 219, please avoid redundancy in the use of "incubation"/"incubated."
Author Response
Reviewer 1
Comments: The manuscript by God et al. describes a study uncovering a novel mechanism of immune evasion in B-cell lymphomas, where tumor-secreted factors impair HLA class II–mediated CD4⁺ T cell activation. Using functional assays and mass spectrometry, the authors identify candidate peptides involved in this process, paving the way for therapeutic intervention. However, the manuscript would benefit from clarification or improvement of a few points, as outlined below:
Responses: We appreciate the reviewer’s thoughtful comments for the improvement of the manuscript. A point-by-point response to the reviewer’s major and minor comments are given below:
Major Points:
Comment #1: In Section 2.3, the authors describe treating cells with conditioned media for 3h prior to antigen pulsing, followed by overnight incubation. It would strengthen the study to investigate whether the observed reduction in antigen presentation is due to HLA downregulation, impaired antigen uptake, or antigen retention—mechanisms already suggested in the second paragraph of the Discussion. It is also essential to demonstrate that the treatment does not compromise cell viability.
Response #1: We totally agree with the reviewer’s comment. We have previously investigated whether the observed reduction in antigen presentation is due to HLA downregulation, impaired antigen uptake, or antigen retention as suggested or discussed in the manuscript. We found that the defect in antigen presentation is not due to alterations of HLA downregulation, impaired antigen uptake, or antigen retention [Immunology. 2014;142(3):492-505. doi: 10.1111/imm.12281, Journal of immunology. 2015;194(4):1434-45. doi: 10.4049/jimmunol.1402382]. Cell viability was not significantly altered by the the treatment as mentioned in those manuscript. However, it is now also included in the revised version of the manuscript.
Comment #2: In Figures 1, 3, and 5, consider including additional readouts of T cell responses, such as IFN-γ production, activation markers (e.g., CD69, CD25), or proliferation assays, to more comprehensively assess functional responses.
Response #2: While IFN-g readout is already included in Figure 1, additional readouts such as IFN-g production for figures 3 and 5 are now included in the revised manuscript as suggested by the reviewer.
Comment #3: Table 1 should include quantitative data and ideally converted into graphical format for clearer visualization.
Response #3: We understand the reviewer’s thoughts. It would be redundant to show some of the findings in Table 1 as similar observations are published in our previous manuscripts ([Immunology. 2014;142(3):492-505. doi: 10.1111/imm.12281, Journal of immunology. 2015;194(4):1434-45. doi: 10.4049/jimmunol.1402382]. Also, it would generate 10 different figures if they are converted into graphical format. Thus, the findings are concisely presented in the Table 1. However, as suggested, some of the data from the Table (e.g., HLA-DR4, HLA-DR7, HLA-DR1 and HLA-A2-restricted presentations) are converted into graphical format for clearer visualization, and included as a supplemental Figure (Figure S1) in the revised manuscript.
Comment #4: Figure 3 currently lacks statistical annotations. Please include appropriate statistical tests and significance indicators.
Response #4: We agree with reviewer. We have re-analyzed the data for statistical significance and modified the graph, and they are now included in the revised version of the manuscript.
Comment #5: Please clarify the technical rationale for the different incubation times used in the co-culture experiments described in Sections 3.1 (in details in 2.3) and 3.5, as this might affect comparability.
Response #5: We appreciate the reviewer’s important point in the methods. For peptide presentation, 6 hrs to overnight incubation time did not make significant differences in our assays. For inhibition assays, preincubation time was 3 hrs and incubation time was at least 6 hrs. These are corrected in the revised manuscript.
Comment #6: Please indicate the statistical tests used in each figure legend, to ensure reproducibility and transparency.
Response #6: The statistical tests used in each figure legend are now indicated as suggested by the reviewer.
Minor Points:
Comment #1: In lines 88–89, there is an open but unclosed bracket, which should be corrected for clarity.
Response #1: The bracket is now closed in the revised manuscript.
Comment #2: In line 209, the word "conformed" appears to be a typo and should likely read "confirmed."
Response #2: We agree with the reviewer. The typo is corrected as “confirmed”.
Comment #3: In line 219, please avoid redundancy in the use of "incubation"/"incubated."
Response #3: We apologize for the unwanted redundance. The indicated redundancy is corrected in the revised version of the manuscript.
Reviewer 2 Report
Comments and Suggestions for Authors
This study is very important and excellent in that it infers that B cell lymphoma secretes previously unrecognized molecules and inhibits HLA class II antigen presentation, thereby making tumor cells invisible to CD4+ T cells and forming an immunosuppressive niche. However, it would be even more interesting if you add the following five points: (1) What are the previously unrecognized secretory molecules of B cell lymphoma? Please explain in more detail. (2) What percentage of each subtype of B cell lymphoma has secretory molecules? (3) Does the presence of secretory molecules lead to a poor prognosis? (4) What is the relationship between the expression of PD-L1 in B cell lymphoma, which has traditionally been considered an immunosuppressive niche? (5) Do other lymphomas and hematopoietic tumors also have secretory molecules? Does the prognosis worsen?
Author Response
Reviewer 2
Comments: This study is very important and excellent in that it infers that B cell lymphoma secretes previously unrecognized molecules and inhibits HLA class II antigen presentation, thereby making tumor cells invisible to CD4+ T cells and forming an immunosuppressive niche. However, it would be even more interesting if you add the following five points: (1) What are the previously unrecognized secretory molecules of B cell lymphoma? Please explain in more detail. (2) What percentage of each subtype of B cell lymphoma has secretory molecules? (3) Does the presence of secretory molecules lead to a poor prognosis? (4) What is the relationship between the expression of PD-L1 in B cell lymphoma, which has traditionally been considered an immunosuppressive niche? (5) Do other lymphomas and hematopoietic tumors also have secretory molecules? Does the prognosis worsen?
Responses: We thank the reviewer for the comment “This study is very important and excellent in that it infers that B cell lymphoma secretes previously unrecognized molecules and inhibits HLA class II antigen presentation, thereby making tumor cells invisible to CD4+ T cells and forming an immunosuppressive niche.”
However, the reviewer suggested that it would be even more interesting if we add the following five points: (1) What are the previously unrecognized secretory molecules of B cell lymphoma? Please explain in more detail. (2) What percentage of each subtype of B cell lymphoma has secretory molecules? (3) Does the presence of secretory molecules lead to a poor prognosis? (4) What is the relationship between the expression of PD-L1 in B cell lymphoma, which has traditionally been considered an immunosuppressive niche? (5) Do other lymphomas and hematopoietic tumors also have secretory molecules? Does the prognosis worsen?
Accordingly, these points are discussed and included in the revised manuscript.
Round 2
Reviewer 1 Report
Comments and Suggestions for Authors
After reviewing the authors' point-by-point response and the revised manuscript, I find that they have adequately addressed the reviewers' comments and have improved the manuscript accordingly.